

# The complete chloroplast genome of *Colobanthus apetalus* (Labill.) Druce: genome organization and comparison with related species

Piotr Androsiuk[1], Jan Paweł Jastrzębski[1], Łukasz Paukszto[1], Adam Okorski[2], Agnieszka Pszczółkowska[2], Katarzyna Joanna Chwedorzewska[3], Justyna Koc[1], Ryszard Górecki[1] and Irena Giełwanowska[1]

[1] Department of Plant Physiology, Genetics and Biotechnology, University of Warmia and Mazury in Olsztyn, Olsztyn, Poland
[2] Department of Entomology, Phytopathology and Molecular Diagnostics, University of Warmia and Mazury in Olsztyn, Olsztyn, Poland
[3] Department of Antarctic Biology, Institute of Biochemistry and Biophysics Polish Academy of Sciences, Warszawa, Poland

Corresponding author
Piotr Androsiuk,
piotr.androsiuk@uwm.edu.pl

## ABSTRACT

*Colobanthus apetalus* is a member of the genus *Colobanthus*, one of the 86 genera of the large family Caryophyllaceae which groups annual and perennial herbs (rarely shrubs) that are widely distributed around the globe, mainly in the Holarctic. The genus *Colobanthus* consists of 25 species, including *Colobanthus quitensis*, an extremophile plant native to the maritime Antarctic. Complete chloroplast (cp) genomes are useful for phylogenetic studies and species identification. In this study, next-generation sequencing (NGS) was used to identify the cp genome of *C. apetalus*. The complete cp genome of *C. apetalus* has the length of 151,228 bp, 36.65% GC content, and a quadripartite structure with a large single copy (LSC) of 83,380 bp and a small single copy (SSC) of 17,206 bp separated by inverted repeats (IRs) of 25,321 bp. The cp genome contains 131 genes, including 112 unique genes and 19 genes which are duplicated in the IRs. The group of 112 unique genes features 73 protein-coding genes, 30 tRNA genes, four rRNA genes and five conserved chloroplast open reading frames (ORFs). A total of 12 forward repeats, 10 palindromic repeats, five reverse repeats and three complementary repeats were detected. In addition, a simple sequence repeat (SSR) analysis revealed 41 (mono-, di-, tri-, tetra-, penta- and hexanucleotide) SSRs, most of which were AT-rich. A detailed comparison of *C. apetalus* and *C. quitensis* cp genomes revealed identical gene content and order. A phylogenetic tree was built based on the sequences of 76 protein-coding genes that are shared by the eleven sequenced representatives of Caryophyllaceae and *C. apetalus,* and it revealed that *C. apetalus* and *C. quitensis* form a clade that is closely related to *Silene* species and *Agrostemma githago*. Moreover, the genus *Silene* appeared as a polymorphic taxon. The results of this study expand our knowledge about the evolution and molecular biology of Caryophyllaceae.

## INTRODUCTION

Chloroplasts are organelles whose main function is the photosynthetic fixation of carbon. They contain the complete enzymatic system for energy production in many metabolic pathways, including the biosynthesis of glucose, amino acids and fatty acids (*Neuhaus & Emes, 2000*). Chloroplasts are uniparentally inherited: they are inherited maternally in most angiosperms and gymnosperms (*Palmer et al., 1988*), but are transmitted in the male line in some gymnosperms (*Sears, 1980*). Their ultrastructure and characteristic genome features indicate that chloroplasts have evolved from free-living cyanobacteria through endosymbiosis (*Gray, 1989*). In a typical terrestrial plant, the chloroplast (cp) genome is a circular DNA molecule with a conserved quadripartite structure composed of a small single copy (SSC), a large single copy (LSC) and two copies of inverted repeat (IR) regions. The cp genome is the smallest of the plant genomes, and it ranges from 120 kb to 165 kb in most species (*Ma et al., 2017*). The variations in size can be attributed mostly to the expansion, contraction or even loss of IRs as well as changes in the length of intergenic spacers (*Palmer et al., 1988*). Due to their compact size, highly conserved status, uniparental inheritance and haploid nature, cp genomes can be effectively used in studies of plant taxonomy and evolution and in species identification (*Kress et al., 2005*; *Chase et al., 2007*; *Parks, Cronn & Liston, 2009*; *Yang et al., 2013*).

The genus *Colobanthus*, a member of the family Caryophyllaceae, consists of 25 species (*The Plant List, 2010*) of tufted, mainly cushion-forming perennials from the Pacific region, Australasia to southern South America, sub-Antarctic islands, maritime Antarctic and Hawaiian mountains. These species are related to *Spergula* and have very small, narrow and dense leaves and solitary, petalless, greenish flowers with four to six, but usually five prominent sepals (*Giełwanowska et al., 2011*; *Alpine Garden Society, 2017*). The available literature is nearly exclusively devoted to only one species of the genus *Colobanthus*, namely *Colobanthus quitensis* (Kunth) Bartl. *Colobanthus quitensis* is a species of special concern as the only representative of Dicotyledoneae in the maritime Antarctic (*Skottsberg, 1954*). *Colobanthus quitensis* has been extensively studied to explore the morphological, physiological and biochemical features that constitute the basis of adaptation to extreme Antarctic conditions (*Bravo et al., 2007*; *Giełwanowska et al., 2011*; *Giełwanowska et al., 2014*; *Bascunan-Godoy et al., 2012*; *Navarrete-Gallegos et al., 2012*; *Pastorczyk, Giełwanowska & Lahuta, 2014*; *Cuba-Díaz et al., 2017*). In contrast, very little is known about the genetic diversity of this species and the genus *Colobanthus* (*Androsiuk et al., 2015*; *Koc et al., in press*). The complete sequence of *C. quitensis* cp genome has been recently published (*Kang et al., 2016*), and it paved the way for more sophisticated genomic studies. However, the significance of those studies will be limited without information about the genome composition of other members of the genus *Colobanthus.* The complete chloroplast genome of eleven Caryophyllaceae species has been sequenced to date (NCBI: the National Center for Biotechnology Information), including eight species of the genus *Silene* (*Silene capitata, S. chalcedonica, S. conica, S. conoidea, S. latifolia, S. noctiflora, S. paradoxa, S. vulgaris*), one species of the genus *Lychnis* (*Lychnis wilfordii*), one species

of the genus *Agrostemma* (*Agrostemma githago*) and one species of the genus *Colobanthus* (*Colobanthus quitensis*).

*Colobanthus apetalus* (Labill.) forms tufts with soft and grassy leaves (1.5-3 cm in length), stems of up to 3 cm, and small greenish flowers (5 mm in diameter) that are more obvious than in other *Colobanthus* species. The sepals often have purple borders, and seeds have low rounded papillae. The species has been recorded in New Zealand, south-eastern Australia, including Tasmania (*Allan, 1961*), and in southern regions of South America (*Skottsberg, 1915*).

In this study, the complete cp genome of *Colobanthus apetalus* was sequenced with the use of the Illumina MiSeq sequencing technology and compared with other Caryophyllaceae species, in particular with *C. quitensis*.

## MATERIALS AND METHODS

### DNA extraction and chloroplast genome sequencing

Fresh leaves of *C. apetalus* were harvested from greenhouse-grown plants (Department of Plant Physiology, Genetics and Biotechnology, University of Warmia and Mazury in Olsztyn). The seeds of *C. apetalus* were collected on the south-eastern shore of Lago Roca, near Lapataia Bay, in the Tierra del Fuego National Park in Argentina. Total genomic DNA was extracted from the fresh tissue of a single plant using the Syngen Plant DNA Mini Kit. The quality of DNA was verified on 1 % (w/v) agarose gel and visualized by staining with 0.5 µg/ml ethidium bromide. The amount and purity of DNA samples were assessed spectrophotometrically.

Genome libraries were prepared from high-quality genomic DNA using the Nextera XT kit (Illumina Inc., San Diego, CA, USA). The prepared libraries were sequenced on the Illumina MiSeq platform (Illumina Inc., San Diego, CA, USA) with a 150 bp paired-end read, version 3.

### Annotation and genome analysis

The quality of the obtained row reads was checked with the FastQC tool. Row reads were trimmed (5 bp of each read end, regions with more than 5% probability of error per base) and mapped to the reference cp genome of *C. quitensis* (NC_028080) using Geneious v.R7 software (*Kearse et al., 2012*) with default medium-low sensitivity settings. The mapped reads were retrieved from the mapping file and used for *de novo* Velvet preassembly (K-mer—23–41, low coverage cut-off—5, minimum contig length—300). Preassembled contigs were extended by mapping row reads using customized settings (minimum sequence overlap of 60 bp and 99% overlap identity) with 30 iterations steps. Elongated contigs were *de novo* assembled after each iteration step to reduce the number and increase the length of sequences and, finally, to create a circular chloroplast genome. The cp genome was annotated using PlasMapper (*Dong et al., 2004*) with manual adjustment. The annotated cp genome was used to draw gene maps with the OrganellarGenome DRAW tool (*Lohse, Drechsel & Bock, 2007*).

## Repeat and SSR analysis

The REPuter program (*Kurtz et al., 2001*) was used to detect and assess genomic repeats, including forward, reverse, palindromic and complementary sequences with a minimal length of 30 bp, Hamming distance of 3, and 90% sequence identity. Chloroplast simple sequence repeats (SSR) or microsatellites were identified in Phobos v.3.3.12 (*Mayer, 2006–2010*) with default settings for perfect SSRs with motif size of one to six nucleotide units. Standard thresholds for the identification of chloroplast SSRs were applied (*Sablok et al., 2015*), i.e., minimum 12 repeat units for mononucleotide SSRs, six repeat units for dinucleotide SSRs, four repeat units for trinucleotide SSRs, and three repeat units for tetra-, penta- and hexanucleotide SSRs. A single IR region was used to eliminate the influence of IR regions, and redundant results in REPuter were deleted manually. The cp genomes of *C. apetalus* and *C. quitensis* were analyzed simultaneously to compare their genomic repeats and identify their SSRs. The NC_028080 sequence downloaded from NCBI represented *C. quitensis* chloroplast genomic data.

## Intra-individual single nucleotide polymorphism

The ''Find Variations/SNPs (Single Nucleotide Polymorphism)'' feature in Geneious software was used to reveal cp genome loci with more than one nucleotide type as candidates for intraspecific SNPs and to assign major and minor alleles. The candidate SNPs were identified on the following conditions: (1) the number of aligned reads for each selected locus must be above 30; (2) the percentage of the minor genotype must be above 10%; (3) maximum variant $p$-value: $10^{-9}$; (4) minimum strand-bias $p$-value: $10^{-9}$.

## Synonymous (Ks) and non-synonymous (Ka) substitution rate analysis

The complete sequence of the *C. apetalus* chloroplast genome was compared with the chloroplast genome sequences of all members of the Caryophyllaceae family currently available in GenBank. The coding sequences of the same protein-coding genes were extracted and aligned separately using MAFFT v7.310 (*Katoh & Standley, 2013*) to estimate synonymous (Ks) and non-synonymous (Ka) substitution rates. The Ks and Ka in the shared genes were estimated in DnaSP (*Rozas et al., 2017*). The analysis was performed in two variants: (1) *C. apetalus* was compared with all representatives of the Caryophyllaceae family (Table1), and (2) the differences in the genes shared by *C. apetalus* and *C. quitensis* were precisely described.

## Phylogenetic analysis

Phylogenetic analyses were performed on 76 sequences of protein-coding genes shared by *C. apetalus* and eleven species belonging to four genera of the family Caryophyllaceae and *Arabidopsis thaliana* (*Sato et al., 1999*) as an outgroup. The appropriate sequences were downloaded from the NCBI database (Table 1). The chosen sequences were aligned in MAFFT v7.310. Bayesian Inference (BI) and Maximum-Likelihood (ML) methods were used for genome-wide phylogenetic analyses in MrBayes v.3.2.6 (*Huelsenbeck & Ronquist, 2001*; *Ronquist & Huelsenbeck, 2003*) and PhyML 3.0 (*Guindon et al., 2010*). Before BI and ML analysis, the best fitting substitution model was searched in Mega 7

**Table 1  GenBank accession numbers and references for cp genomes used in this study.** Species list arranged alphabetically.

| Species | Accession number | Length | Reference |
|---|---|---|---|
| *Agrostemma githago* | NC_023357 | 151,733 bp | *Sloan et al. (2014)* |
| *Arabidopsis thaliana* | NC_000932 | 154,478 bp | *Sato et al. (1999)* |
| *Colobanthus apetalus* | MF687919 | 151,228 bp | in this study |
| *Colobanthus quitensis* | NC_028080 | 151,276 bp | *Kang et al. (2016)* |
| *Lychnis wilfordii* | NC_035225 | 152,320 bp | *Kang, Lee & Kwak (2017)* |
| *Silene capitata* | NC_035226 | 150,224 bp | *Kang, Lee & Kwak (2017)* |
| *Silene chalcedonica* | NC_023359 | 148,081 bp | *Sloan et al. (2014)* |
| *Silene conica* | NC_016729 | 147,208 bp | *Sloan et al. (2012)* |
| *Silene conoidea* | NC_023358 | 147,896 bp | *Sloan et al. (2014)* |
| *Silene latifolia* | NC_016730 | 151,736 bp | *Sloan et al. (2012)* |
| *Silene noctiflora* | NC_016728 | 151,639 bp | *Sloan et al. (2012)* |
| *Silene paradoxa* | NC_023360 | 151,632 bp | *Sloan et al. (2014)* |
| *Silene vulgaris* | NC_016727 | 151,583 bp | *Sloan et al. (2012)* |

(*Kumar, Stecher & Tamura, 2016*), and the model GTR + G + I was selected. A BI partitioning analysis was carried out to develop a majority rule consensus tree with $1 \times 10^7$ generations using the Markov Chain Monte Carlo (MCMC) method. Tree sampling frequency was 1,000 generations. The first 2,500 trees were discarded as burn-in, with a random starting tree. The ML analysis was performed in PhyML 3.0 with 1,000 bootstrap replicates.

# RESULTS

## Genome organization and content

The Illumina MiSeq platform generated around 4,004,432 high-quality reads (∼150 bp in length each, 144.2 bp on average, SD = 19.1; PHRED >30 for 96% of reads) for the *C. apetalus* cp genome, which were then assembled into a complete sequence based on the reference of *C. quitensis* cp genome. The full length of the *C. apetalus* cp genome sequence is 151,228 bp (GenBank accession number MF687919). The genome's circular, quadripartite structure is composed of SSCs with the length of 17,206 bp and LSC with the length of 83,380 bp, separated by a pair of IR elements (IRa and IRb) with the length of 25,321 bp (Fig. 1). The overall GC content of the *C. apetalus* cp genome was 36.65%. The entire *C. apetalus* cp genome contained 131 genes, including 112 unique genes and 19 genes which were duplicated in the IR regions. The group of 112 unique genes consisted of 73 protein-coding genes, 30 transfer RNAs, four ribosomal RNAs and five conserved chloroplast ORFs of various structure and function (*ycf1, ycf2, ycf3, ycf4, ycf68*) (Table 2). Most of the 131 genes in the *C. apetalus* cp genome did not contain introns, 17 contained one intron (*atpF, ndhA, ndhB, petB, petD, rpl16, rpoC1, rps16, trnI-GAU, trnA-UGC, trnA-UGC, trnI-GAU, trnK-UUU, trnG-UCC, trnL-UAA, trnV-UAC, ycf3*), and only 2 contained 2 introns (*rps12* and *clpP1*). The smallest intron was found in the *clpP1* gene (2nd intron, 569 bp), whereas the biggest intron (2,538 bp)—in *trnK-UUU*. The latter was long enough to contain the

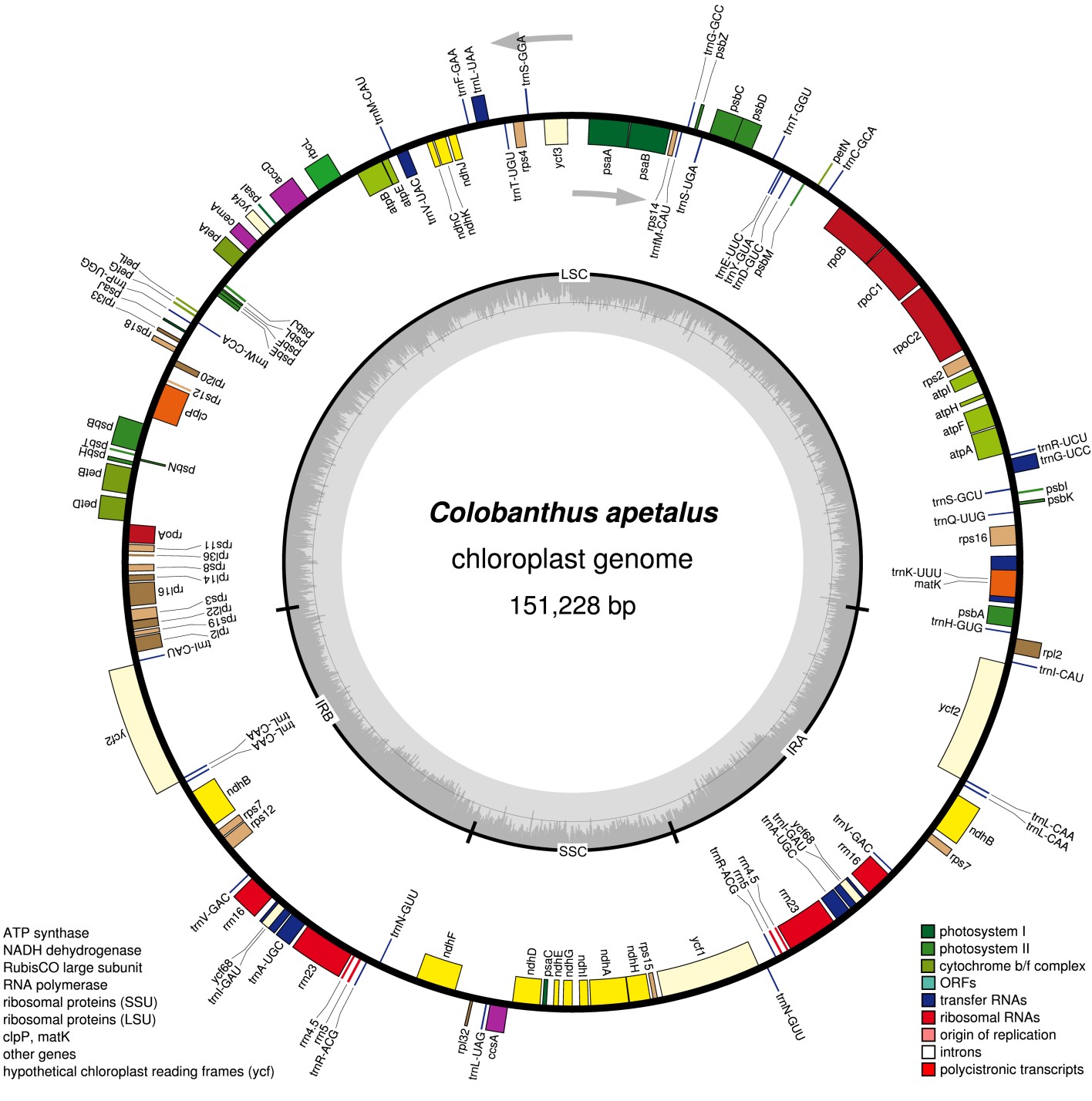

**Figure 1** **Gene map of the *Colobanthus apetalus* chloroplast genome.** Genes drawn inside the circle are transcribed clockwise, and those outside are transcribed counterclockwise (indicated by arrows). Differential functional gene groups are color-coded. GC content variations is shown in the middle circle.

**Table 2  List of genes present in chloroplast genome of *Colobanthus apetalus*.**  Genes list arranged alphabetically.

| Category | Group of gene | Name of genes |
| --- | --- | --- |
| Photosynthesis | Photosystem I | *psaA, psaB, psaC, psaI, psaJ* |
| | Photosystem II | *psbA, psbB, psbC, psbD, psbE, psbF, psbH, psbI, psbJ, psbK, psbL, psbM, psbN, psbT, psbZ* |
| | Cytochrome complex | *petA, petB, petD, petG, petL, petN* |
| | ATP synthase | *atpA, atpB, atpE, atpF, atpH, atpI* |
| | NADH dehydrogenase | *ndhA, ndhB* (x2), *ndhC, ndhD, ndhE, ndhF, ndhG, ndhH, ndhI, ndhJ, ndhK* |
| | Large subunit of RUBISCO | *rbcL* |
| DNA replication and protein synthesis | Ribosomal RNA | *rrn4.5* (x2), *rrn5* (x2), *rrn16* (x2), *rrn23* (x2) |
| | Small subunit ribosomal proteins | *rps2, rps3, rps4, rps7* (x2), *rps8, rps11, rps12* (x2), *rps14, rps15, rps16, rps18, rps19* (x2) |
| | Large subunit ribosomal proteins | *rpl2* (x2), *rpl14, rpl16, rpl20, rpl22, rpl32* (x2), *rpl33, rpl36* |
| | RNA polymerase subunits | *rpoA, rpoB, rpoC1, rpoC2* |
| | Transfer RNA | *trnA-UGC* (x2), *trnC-GCA, trnD-GUC, trnE-UUC, trnF-GAA, trnfM-CAU, trnG-GCC, trnG-UCC, trnH-GUG, trnI-CAU* (x2), *trnI-GAU* (x2), *trnK-UUU, trnL-CAA* (x2), *trnL-UAA, trnL-UAG, trnM-CAU, trnN-GUU* (x2), *trnP-UGG, trnQ-UUG, trnR-ACG* (x2), *trnR-UCU, trnS-GCU, trnS-GGA, trnS-UGA, trnT-GGU, trnT-UGU, trnV-GAC* (x2), *trnV-UAC, trnW-CCA, trnY-GUA* |
| Other genes | Conserved chloroplast ORF | *ycf1* (x2), *ycf2, ycf3* [a], *ycf4* [a], *ycf68* (x2) |
| | Other proteins | *accD, ccsA, cemA, clpP, matK* |

**Notes.**
[a]genes associated with Photosystem I

coding sequence for another gene, *matK*. The *rps12* gene appeared to be trans-spliced because the exon from the 5′end was located in LSC, whereas the remaining two exons from the 3′end of the gene were located in the IR region. Out of the 19 duplicated genes in the IR regions, seven were tRNA, four were rRNA and 8 were protein-coding genes, including three conserved chloroplast ORFs: *ycf2*, *ycf68* and *ycf1* on the border between IRa/IRb and SSC. One *ycf1* from the IRb/SSC border was a functional pseudogene. The SSC region contained 11 protein-coding genes and one tRNA gene, whereas LSC contained 58 protein-coding genes, 22 tRNA genes and 2 conserved chloroplast ORFs (*ycf3* and *ycf4*). The *ycf68* gene was classified as a protein-coding gene in IR; however, according to some authors (*Raubeson et al., 2007*), it probably does not encode a protein.

## Repeat sequences and SSRs
The analysis of repeat sequences from the cp genomes of *C. apetalus* and *C. quitensis* included forward, reverse, palindromic and complementary repeats. A total of 30 repeated sequences with lengths ranging from 30 to 169 bp and sequence identity greater than 90% (Table S1) were identified in *C. apetalus* in the REPuter application. They included 12 forward repeats, 10 palindromic repeats, five reverse repeats and only three complementary repeats. Most of the repeated sequences (23) were dispersed in the intergenic regions (IGS), and only seven were localized within genes. The LSC region was most abundant in repeated

sequences (24), whereas three such elements were found in SSR and IR each. An analysis of the *C. quitensis* cp genome revealed 15 genomic repeats of similar size, from 30 to 168 bp (Table S2). They included seven forward repeats, six palindromic repeats, and two reverse repeats, distributed almost equally in the intergenic regions (8) and genes (7), mostly in LSC (10) and, to a lesser extent, in IR (4) and SSC (1).

The distribution and type of SSRs and microsatellites were also studied in *C. apetalus* and *C. quitensis* cp genomes. Out of the 41 SSRs identified in *C. apetalus*, 33 (80.5%) were located in the LSC region, seven (17.1%) in the SSC region, and one (2.4%) in the IR regions (Fig. 2). The chloroplast SSRs of *C. quitensis* were also identified mainly in the LSC region (79.2%, i.e., 38 SSRs), whereas eight (16.7 %) and two (4.2 %) were located within SSC and IR, respectively. The SSRs can be distributed across three different regions: exons, introns, and intergenic spacers. In the analyzed SSRs from the *C. apetalus* cp genome, 28 (68.3%) repeats were located in the intergenic spacer regions, seven (17.1%) in exons, and six (14.6%) in introns. At this point of the analysis, both species shared a similar pattern of variation where 36 (75.0%) of *C. quitensis* chloroplast SSRs were found in the intergenic spacer regions, five (10.4%) in exons and seven (14.6%) in introns. A more detailed analysis of the SSRs in exons revealed that they were located within the coding sequences of five genes, *rpoC2*, *ndhF*, *ycf1*, *atpA*, and *rrn23*, in both *C. apetalus* and *C. quitensis*. The majority of the microsatellites detected in *C. apetalus* (48.8%) and *C. quitensis* (54.2%) were mononucleotide repeats with one mononucleotide motif (A/T). In di- and trinucleotide SSRs, one microsatellite repeat motif was observed for both species (AT/TA and AAT/TTA, respectively). AAAT/TTTA (25.0%) and AATT/TTAA (25.0%) were the most common tetranucleotide SSRs in both species. In pentanucleotide SSRs, three tandem repeat motifs were observed: AAATT/TTTAA, AAATC/TTTAG and AATCT/TTAGA (Tables S3 and S4).

### Intra-individual single nucleotide polymorphism

A total of four potential interspecies SNPs were detected in intergenic or intronic regions (Table S5). Three were transversions and one was a transition, with minor allele frequency below 12.1%. The detected SNPs were not randomly distributed across the entire cp genome, and they were clustered in LSC region.

### Synonymous and non-synonymous substitution rate analysis

### *C. apetalus* vs members of the Caryophyllaceae family

A total of 76 protein-coding genes from the chloroplast genome of *C. apetalus* were used to analyze synonymous and non-synonymous substitution rates against 11 members of Caryophyllaceae family. Genes with non-applicable (NA) Ka/Ks ratios were changed to 0. The Ka/Ks ratio for most genes was less than 1, with certain exceptions (Table S6A). The Ka/Ks ratio was highest in the *rps7* gene (2.587 for *S. paradoxa* and 2.372 for *S. conica*), whereas it was determined at 1.167 for *S. conoidea* and *S. latifolia*. The Ka/Ks ratio was also high in *rps11* (2.209 for *S. conica*), *rps18* (2.211 for *S. vulgaris*, 1.237 for *S. capitata*), *rps12* (1.416 for *S. paradoxa*, 1.068 for *S. conica*, *S. conoidea*, and *S. latifolia*), *rps16* (1.161 for *S. vulgaris*), *clpP1* (1.314 for *S. chalcedonica*) and *ycf2* (1.245 for *S. conica*, 1.208 for *S.*
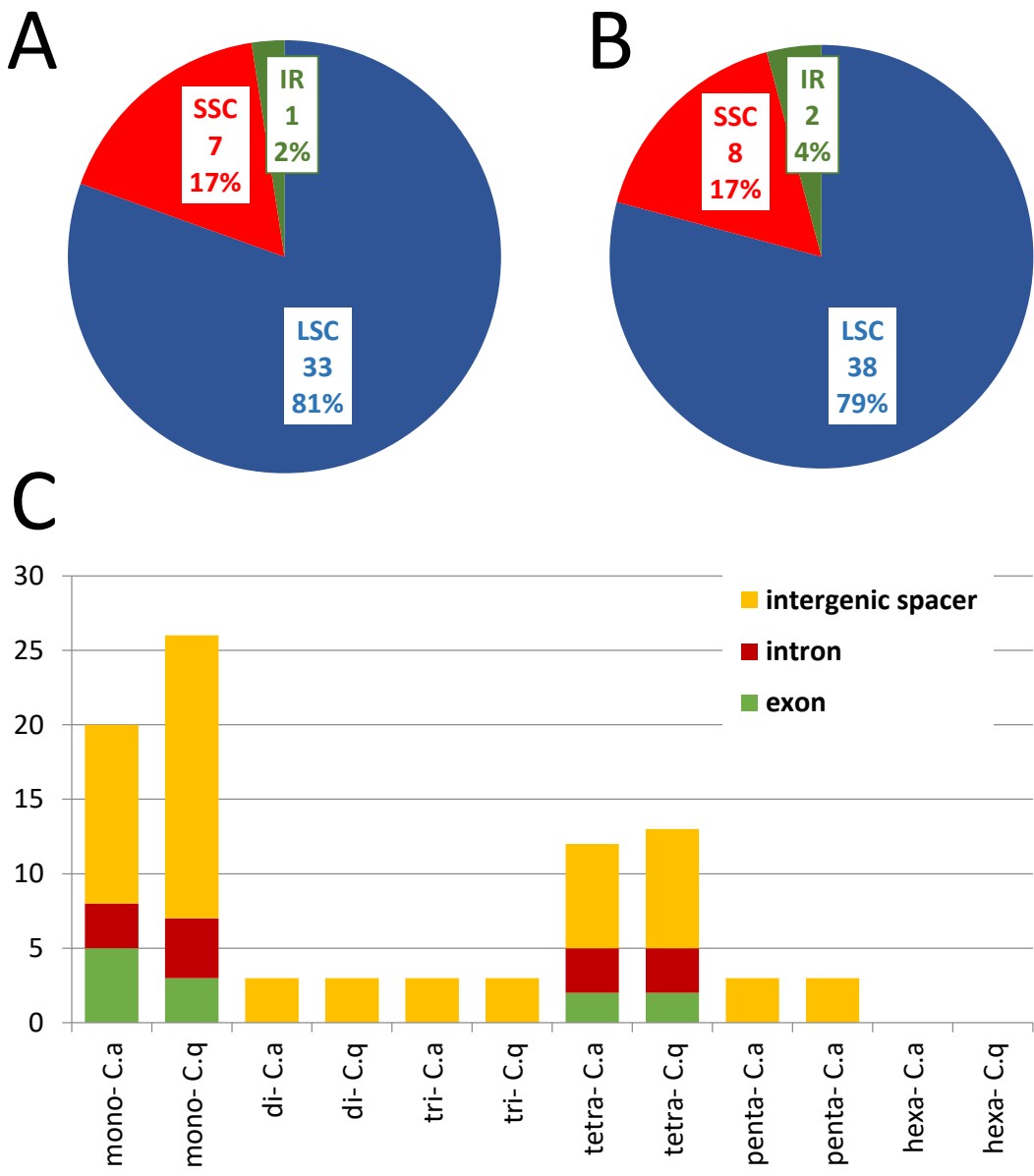

**Figure 2 The distribution, type, and presence of SSRs in cp genome of *Colobanthus apetalus* and *Colobanthus quitensis*.** (A) Presence of SSRs in the LSC, SSC, and IR regions in *C. apetalus* cp genome; (B) presence of SSRs in the LSC, SSC, and IR regions in *C. quitensis* cp genome; (C) presence of SSRs in the protein-coding regions, intergenic spacers, and introns in cp genome of *C. apetalus* (C. a) and *C. quitensis* (C. q).

*paradoxa*, 1.179 for *S. latifolia*, 1.163 *S. conoidea,* and 1.114 for S. *capitata*). A comparative analysis of the genes in each functional group revealed that the substitution rate varied widely across genes, with Ka and Ks values ranging from 0 to 0.757 and from 0 to 0.675, respectively (Table S6A). The highest synonymous substitution rate (average Ks = 0.180) was observed in genes with various functions gathered in the 'other genes' group, whereas the lowest average Ks was noted in genes related to cytochrome b/f complex, photosystem

II and the large subunit of RubisCO (0.106, 0.112, and 0.115, respectively). The highest average non-synonymous (Ka) substitution rate was also observed in the 'other genes' group (average Ka = 0.101), whereas the lowest average Ka was noted in photosystem I and the large subunit of RubisCO (0.006 and 0.007, respectively). Based on Ka/Ks values, 69 genes indicative of purifying selection (Ka/Ks <1) were identified in the analyzed cp genomes. The Ka/Ks ratio was higher than 1.0 in 7 genes in at least one analyzed species, which was indicative of positive selection.

### C. apetalus vs. C. quitensis

More detailed analyses of synonymous and non-synonymous substitution rate were performed for the chloroplast genomes of *C. apetalus* and *C. quitensis* based on 78 protein-coding genes shared by the two species. The analysis revealed additional genes (*accD* and *ycf68*) which were excluded from the previous study because they were not shared by all cp genomes of the family Caryophyllaceae.

The Ka/Ks ratio for all analyzed genes was less than 1 in the range of 0 to 0.388, with the maximum value in *atpH*. However, this value resulted from only one synonymous and one non-synonymous nucleotide substitution in this relatively short (246 bp) sequence. A comparative study of the genes in each functional group revealed minor variations in the substitution rate across genes, with Ka and Ks values in the range of 0 to 0.0084 and 0 to 0.0502, respectively (Table S6B). The highest synonymous substitution rate (average Ks = 0.0162) was observed for the genes related to the large ribosome subunit, whereas a complete absence of synonymous substitutions was noted in the genes related to the cytochrome b/f complex and the large subunit of RubisCO (average Ks = 0). The highest non-synonymous (Ka) substitution rate was observed for the gene of the large subunit of RubisCO (Ka = 0.0028). The average Ka = 0.0021 was noted for the genes with various functions in the 'other genes' group, and the average Ka = 0.0019—for the genes related to the large ribosome subunit. The lowest average non-synonymous (Ka) substitution rate was observed for the genes related to photosystem I and photosystem II (0.0001 for both groups). The noted Ka/Ks values were indicative of purifying selection in all studied genes.

In general, 36 genes had identical sequences (Ka = 0, Ks = 0), whereas the remaining 42 genes showed 99% identity. In 18 analyzed genes, the differences at the nucleotide level were not transmitted to amino acid sequences (Ka = 0). The highest number of nucleotide substitutions was found in *ycf1* and *rpoC2* genes (28 and 15, respectively) which were also characterized by the highest number of non-synonymous amino acid substitutions (15 and 7, respectively). A pairwise alignment of *rps16* in *C. apetalus* and *C. quitensis* demonstrated that in the *C. apetalus* sequence, the substitution of one nucleotide (T → C) in position 201 bp changed the STOP codon (TGA) to CGA encoding arginine (R). The above gave rise to a new "long" allele for *rps16* with 8 additional amino acids (RFKQIKFN). A comparison of cp genomes in *C. apetalus* and *C. quitensis* revealed that the nucleotide substitutions in *rpoB* and *rpl16* genes were accompanied by indel polymorphism. In addition to the substitution of three nucleotides, which was responsible for the shift of two amino acids, the *rpoB* gene encoding the β subunit of RNA polymerase appeared to be shorter in *C. apetalus* due to the absence of one amino acid (glutamine deletion in position 636 of the amino acid
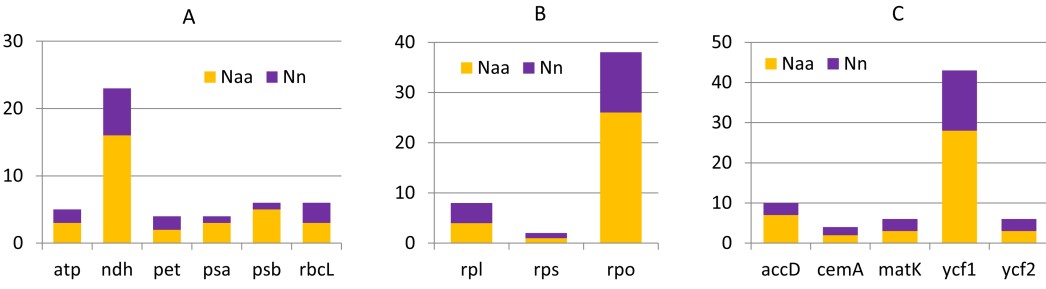

**Figure 3** **Nucleotide and amino acid substitutions in cp genome of *C. apetalus* when compared with *C. quitensis* plastid genome.** $N_{aa}$—number of changed amino acids, $N_n$—number of changed nucleotides in particular group of genes: (A) genes for photosynthesis; (B) self-replication genes; (C) other genes.

**Table 3** **Summary of chloroplast genome characteristic of *C. apetalus* and *C. quitensis*.**

| Genome features | C. apetalus | C. quitensis |
|---|---|---|
| Size (bp) | 151,228 | 151,276 |
| LSC length (bp) | 83,380 | 83,462 |
| SSC length (bp) | 17,206 | 17,208 |
| IR length (bp) | 25,321 | 25,303 |
| Number of genes | 112 | 112 |
| Protein-coding genes | 78 | 78 |
| tRNA genes | 30 | 30 |
| rRNA genes | 4 | 4 |
| Number of genes duplicated in IR | 19 | 19 |
| Overall GC content (%) | 36.65 | 36.7 |

sequence). In the *rpl16* gene in *C. apetalus*, the insertion of thymine in position 396 bp shifted the reading frame and, as a result, the protein sequence was elongated by three additional amino acids (isoleucine, glycine, and threonine) at the 3′-end. A synonymous nucleotide substitution was observed simultaneously within that sequence.

Generally the comparison of cp genomes in *C. apetalus* and *C. quitensis* revealed little or no variation in the genes associated with photosynthesis, whereas information protein genes, including RNA polymerase subunits and ribosomal proteins, underwent greater changes (Fig. 3). Other plastid genes characterized by rapid structural evolution were also identified: *ycf1*, which is essential for cell survival, but whose function has not yet been fully elucidated (*Drescher et al., 2000*), and *accD* which is involved in fatty acid biosynthesis.

## A comparative analysis of the organization of *C. apetalus* and *C. quitensis* chloroplast genomes

A comparison of *C. apetalus* and *C. quitensis* cp genomes revealed considerable similarities in genome composition and size between the species. Both species have the same gene content and order, but *C. apetalus* has a slightly smaller genome (Table 3). A detailed analysis of protein-coding sequences revealed low levels of differentiation and demonstrated that the nucleotide substitution was almost exclusively responsible for the variation, with

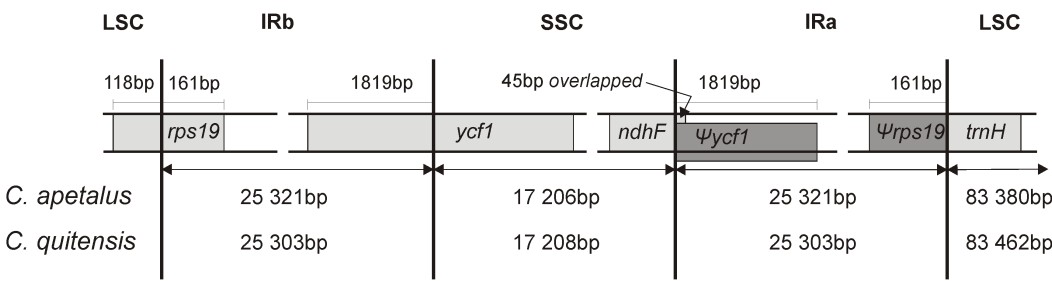

**Figure 4** **Border position of LSC, SSC, and IR regions for *C. apetalus* and *C. quitensis*.** Genes are indicated by boxes and the gaps between the genes and the boundaries are indicated by number of bases unless the gene coincides with the boundary. Extension of the genes are also indicated above the boxes.

scarce indel representation. Therefore, the differences in the length and organization of intergenic spacers were responsible for the observed variations in the size of the cp genome.

The LSC/IRb/SSC/IRa boundary regions of the *C. apetalus* cp genome were also compared to the corresponding regions of *C. quitensis*, and they were found in the same positions within the same genomic elements (Fig. 4). The border between IRa and SSC was located within the coding region of *ycf1*, placing 1,819 bp from the 5′-end within the IRa, which resulted in the Ψ*ycf1* pseudogene in the IRb region. The IRb/SSC border was localized within *ndhF*, and a 45 bp section from its 3′-end overlapped Ψ*ycf1* within IRb. The IRb/LSC border was located within *rps19*, leaving 160 bp from the 5′-end within the IRb region, which resulted in the Ψ*rps19* pseudogene in IRa. The *trnH* gene was located in the LSC region next to the IRa/LSC border.

## Phylogenetic analysis

Both BI and ML methods generated phylogenetic trees with uniform topology. The BI tree revealed that only two nodes had posterior probability values below 1 (0.9978 and 0.9737, respectively). The reconstructed phylogeny revealed that *C. apetalus* and *C. quitensis* formed a monophyletic group that was closely related to a group of *Silene* species and *Agrostemma githago* which formed a solitary branch (Fig. 5). The genus *Silene* appeared as a polymorphic taxon with the two main clades: clade I containing *S. conica*, *S. conoidea*, *S. noctiflora*, *S. latifolia*, *S. vulgaris* and *S. capitate,* and clade II where the only member of the genus *Lychnis* (*L. wilfordii*) was merged with *S. chalcedonica* and *S. paradoxa*.

## DISCUSSION

The *Colobanthus apetalus* chloroplast genome is the second reference quality cp genome to have been sequenced for the genus *Colobanthus* and the 12th reference genome for the family Caryophyllaceae. The *Colobanthus apetalus* cp genome (151,228 bp) is typical in size relative to other angiosperms. It belongs to the group of medium-sized cp genomes in the family Caryophyllaceae: it is only 48 bp smaller than the cp genome of its close relative *Colobanthus quitensis* (151,276 bp), and more than 1,000 bp smaller than the biggest cp genome of *Lychnis wilfordii* (152,320 bp). At the same time, the analyzed genome is more than 4,000 bp bigger than the smallest known cp genome in Caryophyllaceae, which belongs

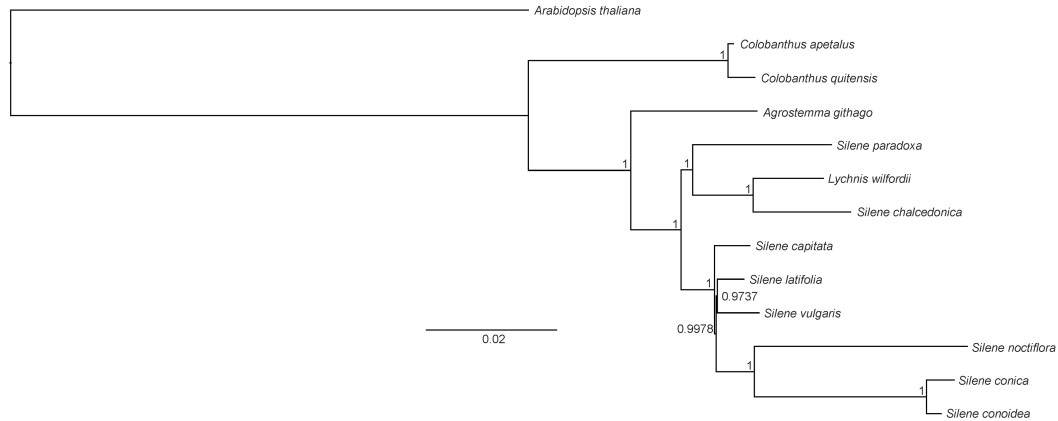

**Figure 5** Phylogeny of *Colobanthus apetalus* and other 11 representatives of Caryophyllaceae based on sequences of sheared 76 protein-coding genes using maximum likelihood method.

to *Silene conica* (147,208 bp). The observed differences in the size of cp genomes result from rearrangements in the genome structure of its non-coding regions, rather than from changes in the number of genes because *S. conica* has 111 genes (protein genes + tRNA genes + rRNA genes) and *L. wilfordii* has 110 genes due to the pseudogenization of the *accD* gene (*Sloan et al., 2012*; *Kang, Lee & Kwak, 2017*). The variations in the size of cp genomes can also be explained by changes in IR structure, such as contractions and expansions (*Raubeson et al., 2007*; *Ravi et al., 2008*; *Wang et al., 2008*). Although the detailed location of IR boundaries changes frequently in angiosperms (*Goulding et al., 1996*), they are generally found within *ycf1* and *rps19* genes. The location of the IR boundary followed the above rule in both analyzed species of the genus *Colobanthus*. Moreover, the IR boundaries appeared to be identical in both species. The IR were very similar in size in *C. apetalus* and *C. quitensis* (25,321 bp and 25,303 bp, respectively), and were located approximately in the middle of the IR size range for the family Caryophyllaceae, with the longest IR in *S. noctiflora* (29,891 bp), and the shortest IR in *S. chalcedonica* (23,540 bp) (*Kang, Lee & Kwak, 2017*). A detailed comparison of cp genomes in Caryophyllaceae (*Kang, Lee & Kwak, 2017*) revealed the coexistence of three types of cp genomes: I) the 'common' type found in most Caryophyllaceae (*Agrosthema githago, Silene capitata, S. latifolia, S. vulgaris* and *Colobanthus quitensis*); II) cp genomes with an inversion of the *ycf3-psaI* region (*S. paradoxa, S. conoidea, S. conica*); III) cp genomes with the most complex structure identified to date among Caryophyllaceae, with several transpositions and/or inversions (*S. noctiflora*). The *C. apetalus* cp genome was highly similar to the *C. quitensis* cp genome, and it was classified as belonging to the first type.

The repeat regions of genomes play an important role in recombination and rearrangement (*Cavalier-Smith, 2002*). In angiosperms, repeat regions are generally found in non-coding regions, and their variations result mainly from illegitimate recombination and slipped-strand mispairing (*Timme et al., 2007*). Similar observations were made in *C. apetalus* where repeats of ≥30 bp were generally found (76.7%) in intergenic regions and introns. The above values are relatively high in comparison with *C. quitensis* (53.3%)

and other Caryophyllaceae, such as *Silene capitata* (56.0%) and *Lychnis wilfordii* (69.2%) (*Kang, Lee & Kwak, 2017*).

Microsatellites (SSRs) are particularly important repetitive elements of the genome. Due to their high reproducibility, ease of scoring, and fast throughput, they are the markers of choice for many population and evolutionary analyses (*Rajwant et al., 2011*). Microsatellite sequences were abundant in the cp genomes of *C. apetalus* and *C. quitensis*, and mononucleotide SSRs were most frequent (48.8% and 54.2%, respectively) with a predominance of the A/T motif. A predominance of A/T in mononucleotide SSRs was previously reported in Caryophyllaceae (*Kang, Lee & Kwak, 2017*), Magnoliaceae (*Kuang et al., 2011*), Poaceae (*Sonah et al., 2011*) and Rhamnaceae (*Ma et al., 2017*). Microsatellites have a unique structure and play an important role in genomic rearrangement and sequence variation, including chloroplast genomes (*Yang et al., 2013*). In our study, most tandem repeats in *C. apetalus* and *C. quitensis* (82.9% and 89.6%, respectively) were found in intergenic spacers and introns which are often divergent hotspots (*Huang et al., 2014*). This observation suggests that these regions can be potentially used to develop new DNA markers for species identification and phylogenetic studies of *Colobanthus* and, possibly, other taxa that are closely related to the family Caryophyllaceae. Other Caryophyllaceae species were characterized by the congruent distribution of SSRs which were also found mainly in non-coding regions (62.7% in *Lychnis wilfordii* and 73.3% in *Silene capitata*) (*Kang, Lee & Kwak, 2017*). Moreover, large numbers of SSRs in *C. apetalus* and *C. quitensis* were found within the coding sequences of only five genes, including *ycf1* which is one of the most rapidly evolving sequences in many groups of plants, including Caryophyllaceae (*Erixon & Oxelman, 2008*; *Sloan et al., 2014*), Campanulaceae, Geraniaceae and Poaceae (*Jansen et al., 2007*).

The second generation high-throughput sequencing technologies generate massive amounts of data with dozens of possible applications, such as the identification of interspecific polymorphism within cp genomes. However, the detection of a polymorphic site within a genome and its separation into major and minor genotype is only the first step in interspecies SNP identification, and further confirmation is required based on extensive sampling. Interspecific polymorphisms point to the heterogeneous nature of the chloroplast population in a given species and could be indicative of heteroplasmy. Heteroplasmy in plastids has been detected in many flowering plants, and it is more common than previously thought (*Tilney-Bassett & Birky Jr, 1981*; *Moon, Kao & Wu, 1987*; *Lee, Blake & Smith, 1988*; *Frey et al., 1999*; *Sabir et al., 2014*). Two mechanisms could be responsible for the development of heteroplasmy in plastids. The more common mechanism which is found in around 20% of angiosperms (*Corriveau & Coleman, 1988*; *Zhang & Liu, 2003*) is a biparental inheritance, where each parent transmits organelles to the zygote. The second mechanism is found in plants with uniparental plastid inheritance, where plastid sorting in parents is incomplete resulting in heteroplasmic gametes. Incomplete sorting could be the mechanism underlying heteroplasmy in the genus Colobanthus. A microscopic analysis of reproductive biology in C. quitensis revealed that the male germ unit is differentiated into a smaller cell containing mainly mitochondria, and bigger one with plastids.

In the process of fertilization in *C. quitensis* only one nucleus of the sperm cell, without cytoplasm fragments of the pollen tube, entered the egg cell, and the proembryo developed according to the Caryophyllad type (*Giełwanowska et al., 2011*). A similar mechanism is likely in *C. apetalus*. *C. quitensis* develops two types of bisexual flowers: opening chasmogamous flowers and closed cleistogamous flowers, where the latter type is favored by low temperatures, high air humidity and strong winds (*Giełwanowska et al., 2011*). The above leads to high selfing rates and the loss of genetic variation in individuals. Considerable similarities in the reproductive biology of *C. quitensis* and *C. apetalus* offer the best explanation for the very small number of SNPs in the *C. apetalus* cp genome, which should be regarded as a minor symptom of heteroplasmy.

Synonymous and non-synonymous nucleotide substitution patterns are a very important element in gene evolution studies (*Kimura, 1983*). Non-synonymous nucleotide substitution is less frequently observed than synonymous substitution (*Makalowski & Boguski, 1998*). In this study, most differences in the chloroplast genes of *C. apetalus* and *C. quitensis* were found in the second and third position of the codon rather than in the first position. However, *ycf1* and *rpoC2* appeared to have the most variable nucleotide sequence and the highest number of non-synonymous substitutions. Similar observations have been made in members of genus *Silene* (Caryophyllaceae), where *ycf1* was one of the most varied coding regions (*Erixon & Oxelman, 2008*; *Sloan et al., 2012*; *Sloan et al., 2014*; *Kang, Lee & Kwak, 2017*). In *rpoB* and *rpl16*, insertion/deletion polymorphisms were also responsible for sequence variation. In addition to three nucleotide substitutions, a comparison of *rpoB* sequences in *C. apetalus* and *C. quitensis* revealed the deletion of one amino acid (glutamine; Q) (KKGQQLLA in *C. quitensis* → KKGQLLA in *C. apetalus*). Glutamine was also deleted from the above amino acid sequence in other Caryophyllaceae. However, an additional conservative substitution of one leucine (L) with isoleucine (I) was noted in the KKGQLLA sequence characteristic for *C. apetalus*, and the new sequence (KKGQILA) was observed in the cp genomes of all Caryophyllaceae that have been sequenced to date. Moreover, the same pattern can be found in more distant relatives of the order Caryophyllales, such as *Spinacia oleracea*, *Dianthus longicalyx* or *Mesembryanthemum crystallinum*. The insertion of an additional nucleotide in the *rpl16* ribosomal protein gene shifted the reading frame and, consequently, added three amino acids (isoleucine, glycine, and threonine) to the gene sequence at the carboxyl terminus of its protein. This type of rearrangement can influence protein structure; however, further research is needed to accurately predict its consequences for protein structure and function (*Berezovsky et al., 1999*). An analysis of *rpl16* in the cp genomes of the sequenced Caryophyllaceae did not produce any evidence to suggest that the gene's length was expanded based on the mechanism observed in *C. apetalus*. The only exception was *Silene paradoxa*, where the C-terminus of *rpl16* was expanded by three, albeit different, amino acids (leucine, glycine, and methionine). Comparative analyses of cp genomes in *Amelopsis, Vitis* and *Liquidambar* also demonstrated high variation within ribosomal protein (*rpl22* and *rps19*) sequences due to a high non-synonymous substitution rate (*Raman & Park, 2016*).

An interesting difference in the sequence of *rpl22* and *rps16* genes was observed in the cp genomes of *C. apetalus* and *C. quitensis*. A comparison of the *rpl22* gene revealed a

substitution of four nucleotides in *C. apetalus*, which led to changes in three amino acids, the appearance of a premature STOP codon and gene contraction by eleven amino acids. A similar 'short' allele of *rpl22* (length of 453 bp = 151 aa) was found in other members of the family Caryophyllaceae (*Agrostemma githago, Lychnis wilfordii, Silene capitata, S. conica, S. conoidea, S. latifolia, S. noctiflora S. paradoxa, S. vulgaris*), and other alleles of the *rpl22* gene were found only in *S. paradoxa* (153 aa) and *S. chalcedonica* (161 aa). In contrast, the substitution of only one nucleotide in *rps16* in *C. apetalus* changed the STOP codon to arginine, which resulted in protein elongation by eight additional amino acids. The 'long' allele of *rps16* is also found in other Caryophyllaceae (*Silene vulgaris, S. noctiflora, S. latifolia, S. chalcedonica, Agrostemma githago*) and Caryophyllales (*Dianthus longicalyx, Spinacia oleracea Mesembryanthemum crystallinum*), but with possible amino acid substitutions. A 'short' allele similar to that found in *C. quitensis* was characteristic also for *Silene conica*. The above examples of unique molecular evolutionary patterns in the genus *Colobanthus* provide valuable inputs for further studies into the evolution and phylogeography of this plant group and the order Caryophyllales. In conclusion, both *C. apetalus* and *C. quitensis* revealed a high degree of sequence conservation in genes that are directly involved in photosynthesis and considerable variations in other genes, including ribosomal proteins and rapidly evolving genes such as *ycf1*. The same conclusions can be drawn from the analysis of synonymous and non-synonymous substitution rates based on the sequences of 76 protein-coding genes shared by all of the studied Caryophyllaceae species. The average Ks values between *C. apetalus* and selected Caryophyllaceae species were estimated at 0.1383, 0.0922 and 0.1789 for the LSC, IR and SSC regions, respectively, with an average Ks of 0.1399 across all regions. The lowest Ks values were found mostly in the IR region, and in two genes (*rpl19* and *ycf1*), the average Ks exceeded the estimate for this group of genes. The LSC region harbored only 12 genes where the average Ks was below 0.0922, whereas no such genes were identified in the SSC region. The distribution of Ks values indicates that the IR region is generally more conserved than LSC and SSC where higher evolution rates are observed. Similar observations have been made by other authors (*Cho et al., 2015*; *Fu et al., 2016*).

The phylogenetic tree presented in this paper was similar to the trees that have been previously developed for Caryophyllaceae based on complete cp genomes (*Kang et al., 2016*) and protein-coding genes (*Kang, Lee & Kwak, 2017*). In all cases, the separate character of the genera *Colobanthus* (so far, represented only by the cp genomes of *C. apetalus* and *C. quitensis*) and *Agrostemma* was established, despite similarities with the heterogeneous genus *Silene* and the genus *Lychnis* nested within it. *Silene* and *Lychnis* were regarded as sister genera within the tribe *Sileneae*; however the taxonomic identities and limitations between these two genera remain unclear (*Lidén, Popp & Oxelman, 2000*). Although some data is available for the genera *Silene* and *Lychnis* (*Greenberg & Donoghue, 2011*; *Kang, Lee & Kwak, 2017*), further research is needed to resolve the relationship between these genera and within the genus *Silene*.

## CONCLUSIONS

The development of a reference cp genome for *C. apetalus* will be valuable for comparative studies of the family Caryophyllaceae and/or the order Caryophyllales which contain halophytic, drought-tolerant and cold-resistant species, including *C. quitensis.* The availability of cp genome sequences for such an interesting group of plants will contribute to the development of new applications in biotechnology, such as chloroplast gene transformation. The reference chloroplast genome is also highly useful for accurate assembly and annotation of cp genomes in other plants within the studied group, identification and analysis of interspecies hybridization events, monitoring their present spread and reconstructing their historical dispersal. The reference cp genome will be particularly useful for monitoring the spread of the genus *Colobanthus* throughout the Southern Hemisphere. The history of the genus *Colobanthus*, its historical dispersal routes and the location of glacial refugia for particular species are fascinating areas of research, but progress in this area is hampered by the lack of sufficient molecular data.

### Funding
This work was supported by internal funding of the Department of Plant Physiology, Genetics and Biotechnology, University of Warmia and Mazury in Olsztyn, and the Polish National Science Centre grant NN303796240. There was no additional external funding received for this study. The funders had no role in study design, data collection and analysis, decision to publish, or preparation of the manuscript.

### Grant Disclosures
The following grant information was disclosed by the authors:
Department of Plant Physiology.
Genetics and Biotechnology.
University of Warmia and Mazury in Olsztyn.
Polish National Science Centre: NN303796240.

### Competing Interests
The authors declare there are no competing interests.

### Author Contributions

- Piotr Androsiuk conceived and designed the experiments, analyzed the data, prepared figures and/or tables, authored or reviewed drafts of the paper approved the final draft.
- Jan Paweł Jastrzębski analyzed the data, contributed reagents/materials/analysis tools, prepared figures and/or tables, authored or reviewed drafts of the paper, approved the final draft.
- Łukasz Paukszto analyzed the data, contributed reagents/materials/analysis tools, authored or reviewed drafts of the paper, approved the final draft.

- Adam Okorski and Agnieszka Pszczółkowska performed the experiments, contributed reagents/materials/analysis tools, authored or reviewed drafts of the paper, approved the final draft.
- Katarzyna Joanna Chwedorzewska, Ryszard Górecki and Irena Giełwanowska contributed reagents/materials/analysis tools, authored or reviewed drafts of the paper, approved the final draft.
- Justyna Koc performed the experiments, authored or reviewed drafts of the paper, approved the final draft.

### Data Availability

Our raw data (complete chloroplast genome sequence for *Colobanthus apetalus*) is available on GenBank, accession number MF687919.

### Supplemental Information

Supplemental information for this article can be found online at http://dx.doi.org/10.7717/peerj.4723#supplemental-information.

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
