# Peer review of "The complete chloroplast genome of Colobanthus apetalus (Labill.) Druce: genome organization and comparison with related species"

_PeerJ, doi:10.7717/peerj.4723_

## Round 0.1 · original submission · Major Revisions

Dear Authors,

I have 2 profound reviews in my hands. One is negative, the other positive. I have decided to give you another chance - hence asking you to perform a major revision of your paper.

Please pay attention especially to the following issues put forward by

Reviewer 1:
(1) .... phylogenetic analysis is not fully described (how was the ML analysis done, exactly?) ....I would recommend removing this part of the manuscript.
(2) The authors could perhaps instead attempt an analysis of RNA editing sites, based on sequence conservation, even in the absence of transcript data.

Reviewer 2:
(1) ....method of genome assembly is not well described
(2) The authors simply counted nt and aa
substitutions, which is least informative and incorrect. The analyses based on e.g. DnaSp or PAML shall be applied.
(3) Did authors observe any evidence for heteroplasmy in their data? This aspect shall be mentioned in Discussion
(4) The authors should clarify the method of genome assembly and perform new analyzes of the Ka/Ks ratios.

Please, also take into account the other comments of the Reviewers.

Sincerely

L. Mucina, Editor

·

Basic reporting

A well-constructed and detailed report of an apparently unremarkable plastid genome from a rather more remarkable plant. The English is clear and easy to follow with only minor and occasional grammatical errors.

Experimental design

The sequencing, assembly and repeat analysis appear to have been carried out thoroughly and I see no obvious issues that need further work. The phylogenetic analysis is not fully described (how was the ML analysis done, exactly?) and brings very little to the paper; the relationship to C. quitensis was never in any doubt, and the other Caryophyllaceae were dealt with better in Kang et al 2017. I would recommend removing this part of the manuscript. The authors could perhaps instead attempt an analysis of RNA editing sites, based on sequence conservation, even in the absence of transcript data. In addition, no attempt appears to have been made to look for any differences in the Colobanthus genome(s) that might play a role in adaptation to the relative extreme conditions under which the plants grow.

Validity of the findings

I have some minor comments on some of the annotations. The authors mention 'five conserved hypothetical chloroplast ORF (open reading frame) of currently unknown functions (ycf1, ycf2, ycf3, ycf4, ycf68)'. This is a very outdated view of these genes' functions. Ycf3 and Ycf4 are known to be assembly factors for Photosystem I and Ycf1 is claimed to be an inner membrane component of a protein import channel (although this is disputed). The ycf68 ORF probably doesn't encode a protein at all (Raubeson et al BMC Genomics 2007).

'...there were genes (ycf1 and accD) for which great acceleration in the rate of
sequence and/or structure evolution in C. apetalus and C. quitensis was observed'. No evidence for acceleration is provided in the manuscript; just because these gene are more divergent than others does not mean that their rate of change has accelerated. They may have always accumulated mutations more rapidly than other genes (indeed this is known to be the case in other lineages).

ClpP is a multisubunit complex, of which the plastid-encoded subunit is just one, so the plastid gene should really be labelled clpP1.

·

Basic reporting

The authors present the complete sequence of the chloroplast genome of Colobanthus apetalus and its comparison with already published chloroplast genome of the Antarctic species Colobanthus quitensis.
Grammar and style are not acceptable, they shall be improved substantially. The authors need to consult a native English speaker. the text is often too lengthy, should be shortened.
The article is well structured with appropriate references. The chloroplast sequence of Colobanthus apetalus has been already published in GenBank.

Experimental design

1. The method of genome assembly is not well described. The authors state:

The obtained high quality short reads were trimmed and then mapped to the reference cp
115 genome of Colobanthus quitensis (NC_028080) using Geneious v.R7 software (Drummond et al
116 2009) with default medium-low sensitivity settings. Contigs were formed by the mapped regions
117 (coverage > 5 reads) located between uncovered gaps. All contigs were mapped using
118 customized settings (minimum sequence overlap of 60 bp and 99 % overlap identity) with 30
119 iterations steps. Next, elongated sequences were de novo assembled to create a circular
120 chloroplast genome.

Which parameters of trimming were used? How was the de novo assembly performed? Were the mapped reads retrieved from the mapping file? Why were there “uncovered gaps” - the corresponding sequence data were missing?
As the reliable sequence is the prerequisite of the following analyses, all the steps leading to the final assembly should be properly clarified.

2. Synonymous and non-synonymous substitution rates
There are well elaborated ways how to analyze Ka/Ks ratio, for example DnaSp (Librado and Rozas, 2009). The authors simply counted nt and aa substitutions, which is least informative and incorrect. The analyses based on e.g. DnaSp or PAML shall be applied.

3. The two Colobanthus species are sister taxa in the phylogenetic tree. The addition of just a single species to the phylogenetic tree published by Kang et al. 2017 does not add very much to our knowledge. The Ka/Ks analysis of the cp proteins across the Caryophyllaceae family would be more contributing.

Validity of the findings

This topic of the manuscript is interesting but it represents only incremental contribution.
There are serious methodical problems described above, which affect the results.

In addition, the authors determined SSR regions. Mononucleotide repeats prevailed. The homopolymer size often varies within the same individual. Did authors observe any evidence for heteroplasmy in their data? This aspect shall be mentioned in Discussion

Additional comments

The authors should clarify the method of genome assembly and perform new analyzes of the Ka/Ks ratios. Sequencing of chloroplast genomes in additional Colobanthus species would improve the impact of this project very much.

---

## Round 0.2 · accepted · Accept

Dear Authors,

I have two, most opposing referee reports in hands: one suggest rejection (Ref. 2), another acceptance (Ref. 1). I tend to disagree with Ref. 2 since this ref. obviously missed the point and requires improvement of phylogeny of Caryophyllaceae, rather than focusing on the topic of the paper itself. I agree that an improved phylogeny would offer a better interpretation tool, yet this was obviously not your aim.
Ref. 1 mentioned some typos etc. still lingering in your manuscript. I suggest that you download Grammarly (it is free on the web) and put your manuscript through the checking paces and fix all remaining typos.

Further, check References again: some punctuation is wrong or missing; please also carefully check using hyphen and en-dash (the latter is used in spans of values, such as page numbers, span of years etc.).
Thank you for considering PeerJ as publishing vehicle of your research.
Sincerely

L. Mucina, Academic Editor

# ·

Basic reporting

There are still some typos and grammatical errors in the text, especially in the newly added sections, but overall the text is clear and easy to understand.

Experimental design

The experimental design is fine as far as it goes - the addition of more genomes might allow more interesting biology to be explored.

Validity of the findings

I believe the findings are valid.

Additional comments

All of the reviewers' criticisms of the initial submission have been satisfactorily addressed, apart from the desire to extend the project to other related genomes.

·

Basic reporting

The revised version of the manuscript was improved, but not substantially. The presented knowledge is incremental. More chloroplast genomes related to Colobanthus are necessary to sequence to achieve the improvement of the phylogenetic tree of Caryophyllaceae.

Experimental design

The calculations based on Ka and Ks were corrected

Validity of the findings

No important conclusions are drawn.

Additional comments

The addition of a single chloroplast genome of the sister species of the previously sequenced and published Colobanthus quitensis does not contribute to our understanding of phylogeny or adaptation in Caryophyllaceae. Please, sequence more plastid genomes, e.g. of Spergula to broaden our knowledge of the evolution in Caryophyllaceae.